# Potential of FTIR-Spectroscopy for Drugs Screening against *Helicobacter pylori*

**DOI:** 10.3390/antibiotics9120897

**Published:** 2020-12-12

**Authors:** Pedro Sousa Sampaio, Cecília R. C. Calado

**Affiliations:** 1DREAMS—Interdisciplinary Center for Development and Research in Environment, Applied Management and Space, Faculty of Engineering, Lusophone University of Humanities and Technologies, Campo Grande, 376, 1749‑024 Lisbon, Portugal; pedro.sampaio@ulusofona.pt; 2CIMOSM—Centro de Investigação em Modelação e Optimização de Sistemas Multifuncionais, ISEL—Instituto Superior de Engenharia de Lisboa, Instituto Politécnico de Lisboa, R. Conselheiro Emídio Navarro 1, 1959-007 Lisboa, Portugal

**Keywords:** drugs screening, Fourier-transform infrared (FTIR) spectroscopy, *Helicobacter pylori*, high-throughput screening

## Abstract

*Helicobacter pylori* colonizes the human stomach of half of the world’s population. The infection if not treated, persists through life, leading to chronic gastric inflammation, that may progress to severe diseases as peptic ulcer, gastric adenocarcinoma, and mucosa-associated lymphoid tissue lymphoma. The first line of treatment, based on 7 to 21 days of two antibiotics associated with a proton pump inhibitor, is, however, already failing most due to patient non-compliance that leads to antibiotic resistance. It is, therefore, urgent to screen for new and more efficient antimicrobials against this bacterium. In this work, Fourier Transform Infrared (FTIR) spectroscopy was evaluated to screen new drugs against *H. pylori,* in rapid (between 1 to 6 h), and high-throughput mode and based on a microliter volume processes in relation to the agar dilution method. The reference *H. pylori* strains 26,695 and J99, were evaluated against a peptide-based antimicrobial and the clinical antibiotic clarithromycin, respectively. After optimization of the assay conditions, as the composition of the incubation mixture, the time of incubation, and spectral pre-processing, it was possible to reproducibly observe the effect of the drug on the bacterial molecular fingerprint as pointed by the spectra principal component analysis. The spectra, obtained from both reference strains, after its incubation with drugs concentrations lower than the MIC, presented peak ratios statistically different (*p* < 0.05) in relation to the bacteria incubated with drugs concentrations equal or higher to the MIC. It was possible to develop a partial least square regression model, enabling to predict from spectra of both bacteria strains, the drug concentration on the assay, with a high correlation coefficient between predicted and experimental data (0.91) and root square error of 40% of the minimum inhibitory concentration. All this points to the high potential of the technique for drug screening against this fastidious growth bacterium.

## 1. Introduction

The Gram-negative bacterium *Helicobacter pylori* persistently colonize the human stomach of half of the world’s population. The elicited immune response due to infection leads to a gastric inflammation that usually, from years to decades, results in 10% of the infected patients in gastric or duodenal ulcers, 1–2% gastric adenocarcinoma, and 0.1% or less gastric MALT lymphoma [1]. There is also evidence linking *H. pylori* infection with some extra-gastric diseases as immune thrombocytopenic purpura, idiopathic sideropenic anemia, and vitamin B12 deficiency [1]. WHO considers this bacterium as a class I carcinogenic agent. The bacteria persistence is due to several mechanisms enabling its evasion to the host’s immune systems, as: is less efficient in inducing innate immune responses in relation to other bacteria; mimics human antigens, or disguises with human antigens; produces diverse virulence factors that will lead to host immune cells division arrest, or reprogramming the immune cells toward a tolerogenic phenotype, and presents a high genetic diversity and plasticity enabling its adaption to assaults of the immune system [1].

*H. pylori* eradication is strongly recommended for patients with the bacteria infection, with a past and present diagnosis of peptic ulcer and gastric cancer, including patients under disease remissions [2]. The bacteria eradication cures the associated gastritis [3], may arrest gastric atrophy, and in some cases may reverse it [4,5], reduces gastric cancer incidence, and is a remarkably effective therapy for low-grade MALT lymphoma of the stomach [6,7].

The recommended first line of treatment is based on a triple-drug therapy, with two antibiotics, clarithromycin and amoxicillin or metronidazole, and one proton pump inhibitor, given twice a day for 7 to 21 days. However, resistance towards these antibiotics is already very high. The efficacy of drug treatment in the 1990s was higher than 90%, where today is much lower, e.g., 55–57% in Western Europe [7,8]. It is expectable that the continuing widespread and imprudent use of antibiotics will lead to continuing increasing antibiotic *H. pylori* resistance. This scenario is enhanced in the case of this bacterium by the high patient non-compliance to the long drugs treatment. It is, therefore, crucial to screen for new anti-bacterial agent’s alternative and more efficient against *H. pylori* in relation to the present ones. More efficient antimicrobials will result in shorter therapies and consequently higher patient compliance and lower rates of antibiotic resistance.

One drawback of searching for antimicrobials against this bacterium is its fastidious growth characteristics. The recommended and conventional tests to estimate the drug’s minimum inhibitory concentrations (MIC) are the E-test and the agar dilution method, which are time-consuming and laborious, besides needing high quantities of the drug to be tested. Both tests are based on 3–4 days of bacteria growth, under microaerophilia at 37 °C, conducted on agar Petri dishes with expensive medium (Hinton as the basal medium) supplemented with blood, whose red opaque color will further difficult the interpretation of results [9,10]. A new high-throughput method that could evaluate the effect of a drug on *H. pylori* growth at a microliter scale and in a sensitive and rapid mode, could strongly contribute to an efficient screening for new alternative drugs. Fourier Transform Infrared (FTIR) spectroscopy is a very attractive technique that could be applied to achieve this goal.

FTIR spectroscopy, in relation to dispersive equipment, increases wavelength accuracy, spectral quality, and reproducibility, making this technique attractive to measure biological processes as in microbiology [11,12]. The evolution of the technique resulted in a highly versatile tool enabling diverse modes of detections (e.g., transmission, transflectance, and attenuated total reflection) which could be associated with fiber optic cables or microscopic analysis. In the present work, a high-throughput analysis conducted in microplates with multi-wells working at transmission mode was applied. The technique is simple to conduct, as usually a minimal sample processing is needed as dehydration, or no pre-processing at all, is rapid as one spectrum is usually taken around one minute, and economic, as no expensive reagents are required. Therefore, the process has been applied in a myriad of microbiological processes, including e.g., prediction of metabolic status along with bacteria cultures [13], to quantify metabolites [14], monitoring infection processes [15], to monitor global responses to stress conditions [16], among other as bacteria typing [11,12].

The IR region is subdivided into three zones: the near-IR (14,000–4000 cm^−1^ or 800–2500 nm) that excite overtone or harmonic bond vibrations; the mid-IR (4000–400 cm^−1^ or 2500–25,000 nm) that excite fundamental bond vibrations; and the far-IR (400–10 cm^−1^ or 25,000–1,000,000 nm), that excited rotational modes. For most of the biological studies, the Mid-IR (MIR) is therefore the region containing higher information concerning vibrational modes of functional groups of biomolecules as proteins, glucose, RNA, and DNA. Therefore, the MIR-spectra theoretically represents the cell’s metabolic fingerprint. Since a specific region of the FT-MIR-spectra may present the contribution of several different bond vibrations, and different biomolecules may present common bonds, a chemometric method, like principal component analysis or partial least squares regression, is usually applied to extract both qualitative and quantitative information.

The present work aims to evaluate the capacity of FT-MIR spectroscopy associated with a microplate reader to retrieve the effect of the drug on the *H. pylori* molecular composition in a rapid (1 to 6 h) and more sensitive mode and using microliter quantities of bacteria cells and drugs, than the conventional technique based on the agar dilution method.

## 2. Materials and Methods

### 2.1. Drug Minimum Inhibitory Concentration (MIC), Determined by the Agar Dilution Method

The MIC of antimicrobials against *H. pylori* 26,695 and J99 strains were determined using *H. pylori* selective media (Biogerm, Maia, Portugal), according to EUCAST [17]. An antimicrobial compound, peptide-based, that is under development in the present laboratory was used. Due to patent issues, the peptide sequence is not disclosed. The following concentrations of the peptide-based drug were tested: 0, 0.78, 1.56, 3.12, 6.25, 12.5, 25, 50, 75 and 100 mg/L. The following concentration of the clinical antibiotic clarithromycin was also tested: 0, 1.5, 3.75, 7.5, 15, 30, and 75 mg/L. The Petri dishes were incubated for 72 h at 37 °C in microaerophilic conditions established by using an anaerobic jar and microaerophilic sachets (CampyGen, Oxoid, Hampshire, UK). The assays were conducted in duplicate.

### 2.2. Incubation of H. pylori with the Drugs for FT-MIR-Spectra Acquisition

Four sets of experiments were performed with the 26,695 *H. pylori* strain representing different incubation conditions of the bacterium with a peptide-based antimicrobial.

*H. pylori* 26,695 or J99 was grown for 48 h on *H. pylori* selective media (Biogerm, Maia, Portugal) at 37 °C in microaerophilic conditions (CampyGen, Oxoid, Hampshire, UK). The biomass was collected from the Petri dishes and suspended in phosphate buffer (pH 7.0 50 mM) to an optical density (O.D.) (at 600 nm) of 2.0 or 4.0 (Synergy 2 with the GEN5 software, (BioTek Instruments, Winooski, VT, USA).

The 26,695 *H. pylori* was incubated with a peptide-based drug in mixtures as described in Table 1. Each experiment was conducted in triplicate.

The *H. pylori* J99 strain was incubated, at a DO of 2.0 and 5% (*w/v*) of brucella broth, with the following concentrations of clarithromycin: 0, 1.5, 3.75, 7.5, 15, 30, and 75 mg/L. The experiments were conducted in quadruplicate.

### 2.3. FTIR-Spectroscopic Analysis

25 µL of the incubation mixture with the bacteria and antimicrobials was placed on IR-transparent Si-microtiter plates with 96 wells (Bruker Optics, Ettlingen, Germany) and dehydrated for 2.5 h using a vacuum desiccator (ME2, Vacuubrand, Wertheim, Germany). The FTIR spectra were acquired using an HTS-XT associated with a Vertex-70 spectrometer (Bruker Optics, Ettlingen, Germany) at a resolution of 2 cm^−1^ between 400 to 4000 cm^−1^. Each final spectrum results from the average of 40 spectra acquisitions.

### 2.4. Chemometrics Analysis

The following pre-processing techniques were applied to the whole FTIR spectra: Baseline correction by subtracting a constant value to the original spectra, multiplicative scatter correction (MSC), and the first spectral derivative based on a Savitzky–Golay filter [18]. If not otherwise stated, the Savitzky–Golay filter was based on a fourth level polynomial and 15 points.

Following the pre-processing, a principal component analysis (PCA) and the partial least squares (PLS) regression were employed. The number of latent variables (LV) to be incorporated into the PLS models was chosen based on the root mean squared prediction error (RMSPE) and the correlation coefficient between the experimental and predicted drug concentration. PLS regression models were built on independent data sets with 2/3 of data (i.e., for model calibration), while the remaining 1/3 of data was used for model validation [19].

Pre-processing and PCA were conducted with MATLAB (MathWorks, Natick, MA, USA), while PLS was conducted by The Unscrambler X, version 10.5 (CAMO Software AS, Oslo, Norway). ANOVA was conducted by Microsoft Excel.

## 3. Results and Discussion

The first set of experiments was conducted with the reference *H. pylori* 26,695 strain and an antimicrobial peptide-based. This strain was the first *H. pylori* bacteria to be fully sequenced in 1997. The peptide-based drug was in an aqueous solution and it is intended that the antimicrobial effect be achieved directly on the stomach after its oral administration. Usually, the antibiotics therapy against *H. pylori* infection is associated with a proton pump inhibitor to increase stomach pH [1]. The minimum inhibitory concentration (MIC), based on the conventional agar dilution method, obtained for this drug was 100 mg/L.

As the FTIR spectra reflect the bacteria cell molecular composition in a highly specific and sensitive mode, to develop a reproducible assay it is necessary to standardize the bacteria growth conditions used before the assay, the composition of the incubation mixture of the assay, the assay incubation time and spectral processing procedures [13].

The bacteria were grown using a standard protocol based on 48 h culture on selective media at 37 °C in microaerophilic conditions. This growth condition will maximize bacteria viability while minimizing the appearance of dormant and non-culturable coccoid forms. It was also observed that a cell quantity with an O.D. between 2.0 and 4.0 should be used to obtain an FTIR spectrum with a good signal-to-noise ratio as represented in Figure 1A.

The media composition affects the bacteria’s susceptibility to a defined drug. In general, cells are more sensitive to environmental conditions that promote high metabolic activities. Therefore, to potentiate the effect of the drug on the cell metabolism, the bacteria were incubated with Brucella broth, as it contains sodium chloride, sodium bisulfite, glucose, and as a complex nitrogen source, with peptic digest meat and yeast extract. The incubation mixture, containing the bacteria, antimicrobial, and brucella broth, was maintained between 1 to 6 h at 37 °C at normal atmosphere before sample dehydration and spectra acquisition.

The spectra pre-processing was also optimized. The spectra baseline correction, which aims to minimize dissimilarity between spectra due to baseline deviation, was conducted by subtracting to the original spectra a constant value (i.e., by off-set correction). The FTIR spectra collected from the dry film can significantly be influenced by several physical effects such as light scattering originated from particles of different sizes and shapes or in homogeneities in the sample distributions [18]. To minimize these effects on spectral data, after the spectra off-set correction, a multiplicative scatters correction (MSC) and the first spectral derivative were applied. The effect of the pre-processing techniques was evaluated on triplicated experiments on the score-plot of a principal component analysis (PCA). It was observed that the application of MSC minimizes substantially the variance between spectra replicates, evidencing the strong contribution of physical distortions on the spectra (Figure 1A vs. Figure 1B). It is observed that after application of the MSC, data from experimental triplicates are close together in the PCA score plot (Figure 1C vs. Figure 1D). Indeed, the variance within replicates of PCA data decreased significantly after MSC application (Table 2).

The spectra first derivative was also evaluated, due to its general use to resolve bands overlapping. It is also mostly used for baseline shift correction as the first derivative of a constant is zero. However, two major drawbacks of applying derivatives are the reduction of the signal scale and noise amplification. As deriving a spectrum corresponds to multiplying a Fourier-Transform by 2πiν, the noise amplification increases with the radiation frequency (ν) and the order of the derivative. One method of avoiding noise amplification is smoothing the spectrum with filters as the Savitzky–Golay filter. After testing several filters, the best smoothing was conducted with a Savitzky–Golay filter with order 4 and a window size of 15, as it smoothed the noise while preserving small peaks (Figure 2A,B). It was observed that the application of the first derivative enables to increase the data set variance capture by the PCA. For example, for the experiment represented in Figure 2, the PC1 and PC2 without the application of the first derivative represented 91.9% of the data set variance, where the application of the first derivative allowed the capture by PC1 and PC2 of 96.7% of the data set variance. On that experiment, the replicate variance also slightly decreased, as the ratio between the variance captured in the replicates of each sample and the variance of the whole dataset decreased from 0.310 to 0.044 after the first derivative application (Figure 2C,D).

Figure 3 represents the PCA score-plots of different incubation mixtures, conducted with different quantities of bacteria, drug, brucella broth, and submitted for different incubation periods. All the spectra were pre-processed by off-set and MSC correction. The graphs on the right were further pre-processed with the first derivative in relation to the equivalent graphs on the left. It was observed, in all of the cases, that sample triplicates, representing the effect of a defined drug concentration, tend to group and, apart from the other samples treated with other drugs concentrations, indicating that FTIR spectra captured the effect of the drug on the bacteria cell metabolism and in a reproducible mode. It was also observed that even for the lowest drug concentration tested (equivalent to 2.5% of the MIC) that samples grouped together and apart from the control sample conducted without the drug. Therefore, the described methodology enables one to detect the impact of the drug on the cell metabolism in a more sensible mode than the conventional assay based on the agar dilution method, according to other authors observations of the high sensitivity of the technique to infer the impact of culture conditions on the bacteria metabolism [13,14,16].

In summary, in this first experiment conducted with 25% of brucella broth and a cell quantity equivalent to an O.D. of 4.0, it was observed that samples treated for 1h with 100 mg/L of the drug, were grouped together and apart from other conditions (Figure 3A,E).

The second set of experiments were conducted with lower quantities of bacterial cells (with an O.D. of 2.0 instead of 4.0) and brucella broth (with 5% instead of 25% (*w/v*) (experiment 2, Table 1 and Figure 3B,F). It was observed that the control sample, without any drug, is no longer so well apart from other conditions with the drug, as observed in experiment 1 (Figure 3E vs. 3F). Furthermore, the samples treated with 100 mg/L of the drug are no longer so well separated from the samples with lower drug concentrations as observed in experiment 1. Only for drug concentrations higher than 2× MIC, was possible to observe sample separation on the PCA score-plot in relation to samples treated with lower drug concentrations (Figure 3B,F). This lower sensitivity is probably due to the assay’s lower nutrient content.

To increase the effectiveness of the drug on cell metabolism, the incubation period was increased. With 3 h of incubation (experiment 3). However, the assay was still not as sensible as in experiment 1. After 6 h of incubation (experiment 4) it was observed a clear spatial separation of samples treated with drugs concentrations higher or equal to the MIC in relation to the other conditions (Figure 3A,H). Furthermore, using this incubation condition (with lower nutrient content and a high incubation period) the reproducibility was increased, as sample replicates are closer, being also possible to observe the higher spatial separation of the control samples (without the drug) from all the other samples in relation to experiment 1. With those incubation conditions, it was also possible to capture by the PC1 and PC2 the highest data set variance in relation to all the other conditions evaluated (Figure 3). Experiment 4, represents, therefore, the best conditions to reproducibly capture the effect of the drug on the cell molecular fingerprint.

A partial least square (PLS) regression model was build based on bacteria first derivative spectra of experiment 4. A very good model, based on 3 latent variables, was achieved, enabling one to predict from the bacteria spectra, the drug concentration in the assay. The model presented, for the validation data set, a correlation coefficient of 0.93 and a root mean square error of prediction of 34 mg/L (Figure 4). The PLS regression vector obtained highlighted the following FTIR spectra regions that presented the highest correlation with the drug effect on the bacteria metabolism: 1000 to 1800 cm^−1^, 2800 to 3000 cm^−1,^ and 3200 to 3400 cm^−1^ (Figure 4B). The regions between 1500 and 1800 cm^−1^ are mainly due to amide I and amide II adsorptions, and usually are the most prominent bands in a bacterial infrared spectrum since proteins tend to be, on average, half of bacterial cell composition. The peak at 1738 cm^−1^, is assigned to the ester C=O stretching of the phospholipids. In general, at 1080 cm^−1^ and 1236 cm^−1^ there is a strong contribution of the P=O bond present in phosphate groups of molecules as nucleotides as ATP and nucleic acids, phosphorylated proteins, and lipids. Between 1000 and 1100 cm^−1^ a general contribution from C–O, C–C, C–OH, C–O–C bonds present in carbohydrates occurs. The region between 2800 and 3000 cm^−1^ presents a high contribution of stretching vibrations of the C-H bonds from lipids. The contribution between 3200 to 3400 cm^−1^ are from stretching vibrations of the N-H bond in amide-A functional group of proteins (that adsorbs around 3200 cm^−1^), and from stretching vibrations of O–H of hydroxyl groups (that adsorbs around 3500 cm^−1^).

Ratios of absorbance between peak heights from the bacteria spectra were determined. The peaks were selected according to the regions highlighted by the PLS regression vector (Figure 4B). The ratios of peak heights were considered instead of isolated peaks, as this will minimize the effect of physical interferences between spectra, highlighting the chemical characteristics of the spectrum. The ratios were selected also according to other author’s analyses [15,20]. Table 3 presents the fifteen peak ratios analyzed, were “A” followed by the number represents the absorbance at that wavenumber. It was evaluated if the variance of the mean of the peak’s ratios of bacteria incubated with drugs concentrations lower than the MIC (i.e., with 0, 2.5, 5, 15, 50, and 75 mg/L) were significantly different from bacteria incubated with drugs concentrations higher or equal to the MIC (i.e., with 100, 200 and 400 mg/L). From the fifteen ratios evaluated, the following seven ratios were statistically different, in mean terms, at 5% significance: A2879/A2852, A1743/A2852, A1743/A1548, A1656/A1548, A1457/A1548, A1241/A1170, and A1241/A1548 (Table 3, Figure 5). These significant ratios highlight the impact of the drugs on the bacteria metabolism, as on the protein synthesis (indicated by the ratio between amide I and II, i.e., A1656/A1548, and diverse ratios in relation to amide I peak) and on the cells membrane composition as pointed by the ratio CH_3_/CH_2_ (i.e., A2879/A2852) and the ratios including the peak at 1743 cm^−1^ and 1170 c^−1^ from carbonyl vibrations of esters of phospholipids and at 2879, 2852, and 1457 cm^−1^ from methyl and methylene groups. Figure 5 represents, the average value of some of these bacteria spectral peak ratios, along with diverse drugs concentration, being observed that on average A2879/A2852, A1743/A1656, and A1241/A1170 were higher for drugs concentrations equal or higher to the MIC.

The present method was subsequently evaluated against the reference *H. pylori* J99 strain. The reference strains J99 and 26,695 are the most studied and characterized *H. pylori* strains [1] and were originally isolated from an American patient with peptide ulcer and an English patient with chronic gastritis, respectively. These strains were the first ones to be fully sequenced. The condition previously optimized with the 26,695 *H. pylori* strain was tested with the J99 strain. Interestingly, spectral data from the J99 *H. pylori* strain (with and without an antimicrobial) was in a different cluster on the PCA score plot in relation to spectra from the 2669 *H. pylori* strain (Figure 6A). This is according to the fact that FTIR spectroscopy can acquire the molecular fingerprint of a bacteria with high sensitivity and specificity, enabling its application for bacteria typing at species and strain level as conducted for example to lactic acid bacteria [21], *Staphylococcus aureus* strains [22], *Klebsiella pneumoniae* strains [23] *Acitenobacter baumannii* clones [24] among others [11,12,25].

In preliminary work, FTIR spectra of six different strains of *H. pylori* presented data clusters in the PCA score plot (Figure 6B). Each color on Figure 6B represents replicated growth of the same strain, being observed that data from the same strain grouped together and apart from the other five strains. As all these strains were grown in the same conditions (i.e., based on 48 h on selective media at 37 °C in microaerophilic conditions), the spectral differences most probably reflect the different molecular composition between strains. This points out that FTIR spectroscopy captures the molecular fingerprint at a strain level of *H. pylori*, as previously observed with other bacteria [11,12,22,23,24,25]. This will not impact drug screening, as usually, for a defined bacterium species a reference strain is used.

*H. pylori* J99 was submitted to clarithromycin, one of the most common antibiotics applied against *H. pylori* infection. This antibiotic MIC, against J99 *H. pylori,* was 15 mg/L. The PCA score-plot shown in Figure 7 represents quadruplicate experiments, where blue symbols correspond to bacteria incubations with a drug concentration below the MIC, the red symbols to the MIC, and the remaining colors (green and pink) to values above the MIC. Diverse data from replicated experiments were superimposed. All this reinforces that FTIR spectroscopy captured the effect of the concentration of the drug on the bacteria molecular profile in a reproducible mode and with high specificity and sensitivity.

The PLS regression model developed for the J99 strain enables to predict, from the bacteria spectra, the assay clarithromycin concentration. The model, based on 6 latent variables, presented for the validation data set a 0.97 correlation coefficient between predicted versus experimental drug concentration and a root mean square error of 2.1 mg/L, corresponding to 14% of the MIC (data not shown). In summary, for each *H. pylori* reference strain evaluated (26995 and J99), FTIR spectroscopy enabled to capture in a reproducible mode the effect of antimicrobial concentrations on the bacterial molecular fingerprint.

The J99 *H. pylori* spectra presented ratios of peaks that in mean terms were statistically different, at 5% significance, between bacteria incubated with clarithromycin concentrations below the MIC and bacteria incubated with clarithromycin concentrations equal or higher to the MIC (Table 4). The peak ratios that were statistically different for the J99 strain were different from the ones observed with the 26,695 strain. For example, the ratios including the ester band at 1743 cm^−1^ were significantly different for the 26,995 strain but not for the J99 strain. Common ratios that were statistically different with both strains were e.g., the ratio between the amide I and amide II (A1656/A1548), the ratio between phosphate groups at 1241 cm^−1,^ and the ester band at 1170 cm^−1^. These differences most probably reflect the high sensitivity and specificity of FTIR spectroscopy in capturing the bacterial molecular composition at the strain level.

A drug screening assay based on the present method would be conducted on a reference *H. pylori* strain, and the effect of the drug on the cell’s metabolism would be monitored by its impact on spectral ratios (as presented in Table 3 and Table 4) or by PLS regression models valid for diverse drugs. For that, another PLS regression model was developed based on spectral data from both reference strains (26,695 and J99) tested against the two antimicrobials (the peptide-based drug and clarithromycin), that included: replicates of 10 different conditions with bacteria incubated with drugs concentrations that, in a conventional method, did not inhibit the bacteria growth; Replicates of 6 different conditions with bacteria incubated with drugs concentrations that, in a conventional method, inhibited the bacteria growth. With replicas, a total of 55 conditions were considered: n = 36 for the model building, i.e., model calibration, and an independent dataset of n = 19 for model validation. The PLS regression model, based on 7 latent variables, resulted in the validation data set on a 0.91 correlation coefficient and a root mean square error of 40% in relation to the MIC of the drug (Figure 8). Usually, a drug screening assay only needs to predict if the drug will limit bacteria growth. The possibility of the assay to predict the MICs (even with a high root mean square error of 40%) enables the prediction of the effect of the drug on the bacteria with the extra benefit of pointing to the drug’s MICs.

## 4. Conclusions

*H. pylori* is a fastidious growth bacterium that infects half of the world population, being the major risk of severe gastric diseases as peptic ulcers and gastric cancer. In the present work, a reproducible method enabling to predict the impact of the drug on *H. pylori* growth was defined. The method is based on a rapid (between 1 to 6 h) and high-throughput (with 96-wells microplates) FTIR spectral analysis of a microliter volume suspension of the bacteria incubated with the drug. The method enables to predict the impact of the drug on the bacteria growth, from e.g., the analysis of the spectra peak ratios or from PLS regression models. The method presents therefore a high potential for drug screening over this critical and fastidious growth bacterium.

## Figures and Tables

**Figure 1 antibiotics-09-00897-f001:**
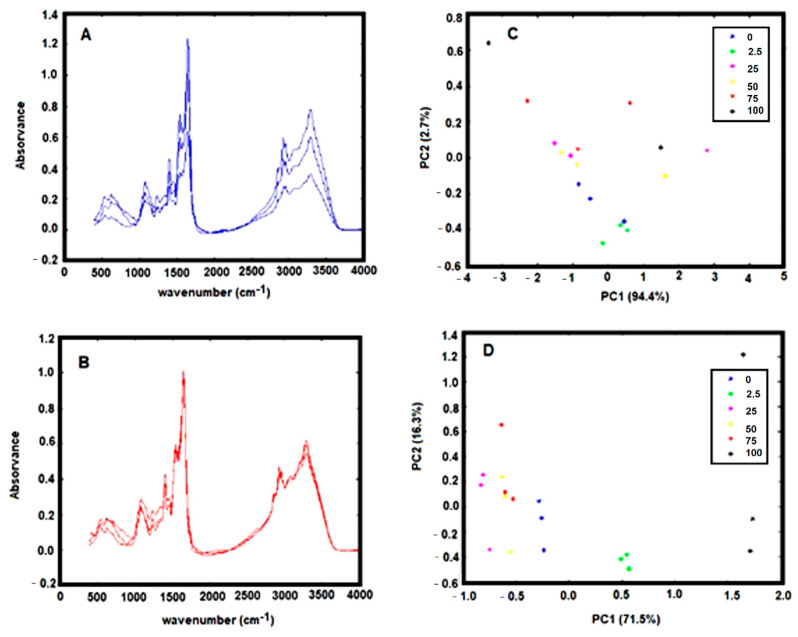
(**A**) Triplicate FTIR spectra, of dehydrated *H. pylori* with an OD 600 nm of 4.0, baseline-corrected by off-set. (**B**) Same replicate spectra as indicated in A with a multiplicative scatter correction (MSC); (**C**) PCA score-plot relative to FTIR spectra, of *H. pylori* incubated with different concentrations of the drug model, baseline-corrected by off-set. (**D**) Same data as presented in C with MSC correction. Legend of symbols: Blue—0 mg/L; Green—2.5 mg/L; Magenta—25 mg/L; Yellow—50 mg/L; Red—75 mg/L; and Black—100 mg/L.

**Figure 2 antibiotics-09-00897-f002:**
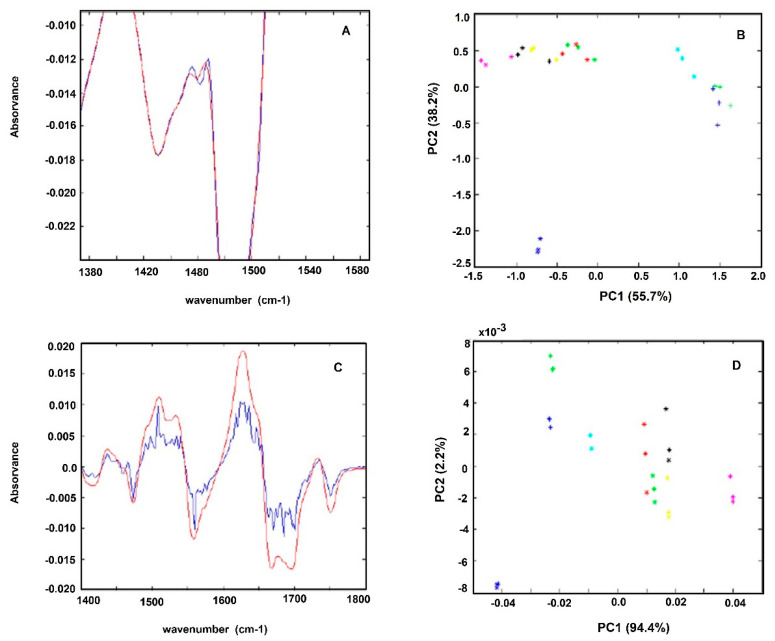
Effect of application of Savitzky–Golay filter on the: (**A**) FT-MIR spectra; (**B**) first derivative spectra (**B**). Only some regions of the spectra where the effect of the filter is high are shown. Effect on the PCA analysis using a: (**C**) normal spectral; (**D**) first-derivative spectra with a Savitzky–Golay filter. Legend of symbols: Blue—0 mg/L; Green—2.5 mg/L; Magenta—5 mg/L; Yellow—25 mg/L; Red—50 mg/L; Black—75 mg/L; Cyan—100 mg/L; Blue-cross—200 mg/L; Green (cross)—400 mg/L.

**Figure 3 antibiotics-09-00897-f003:**
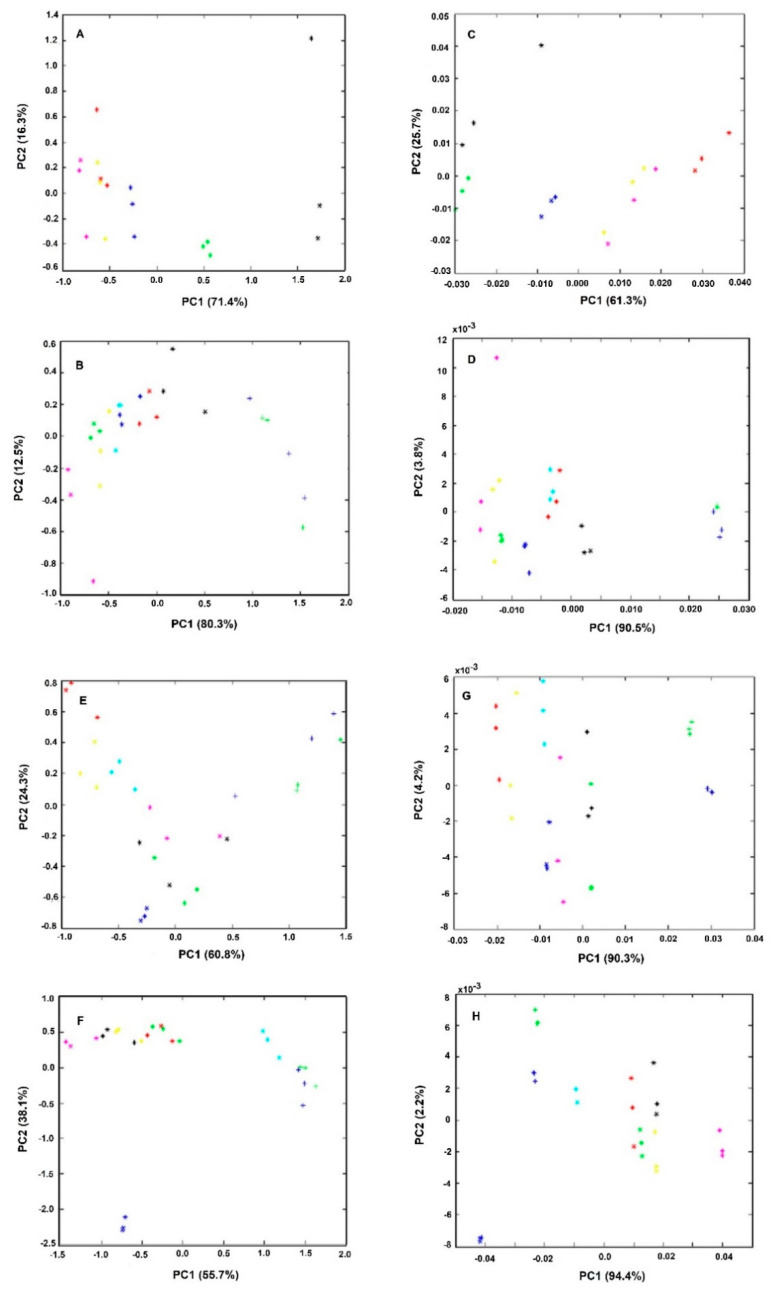
PCA of triplicates of bacteria spectra obtained from experiments as described in Table 1. All spectra were pre-processed by off-set and MSC correction. The graphs on the right (**E**–**H**) were further preprocessed by the first spectral derivative in relation to the respective left graphs (**A**–**D**). Legend of symbols: Blue—0 mg/L; Green—2.5 mg/L; Magenta—5 mg/L; Yellow—25 mg/L; Red—50 mg/L; Black—75 mg/L; Cyan—100 mg/L; Blue-cross—200 mg/L; Green (cross)—400 mg/L.

**Figure 4 antibiotics-09-00897-f004:**
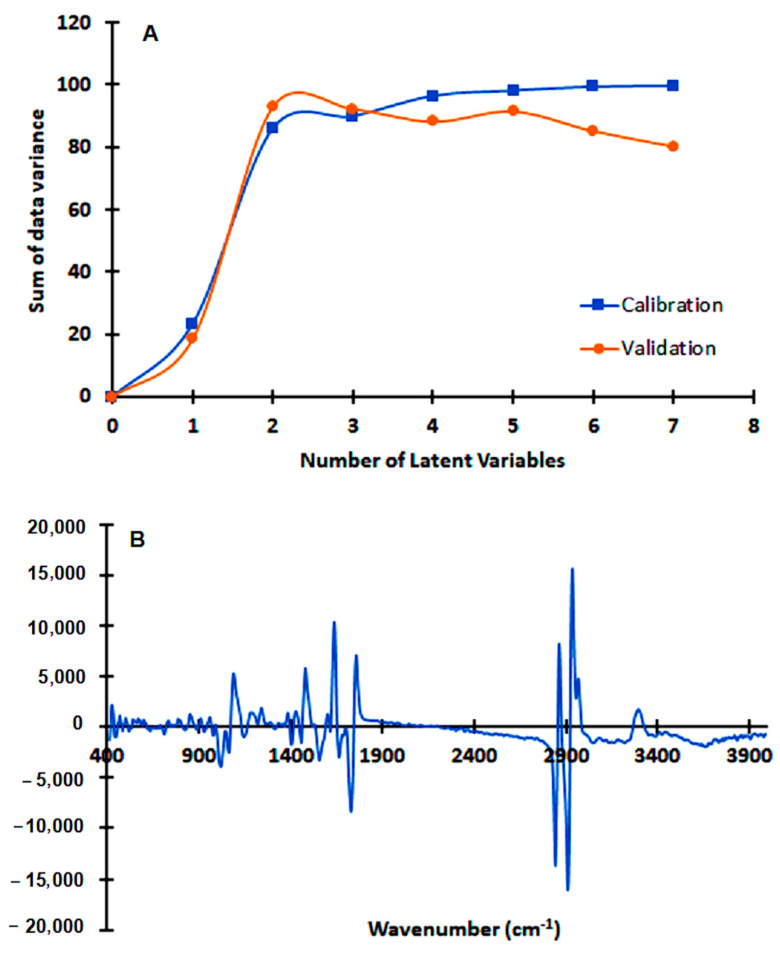
A partial least square (PLS) regression model based on bacteria spectra of experiment 4 (Table 1). Spectra were pre-processed by off-set, MSC, and first derivative. (**A**) The sum of data variance along with latent variables; (**B**) Regression vector with three latent variables.

**Figure 5 antibiotics-09-00897-f005:**
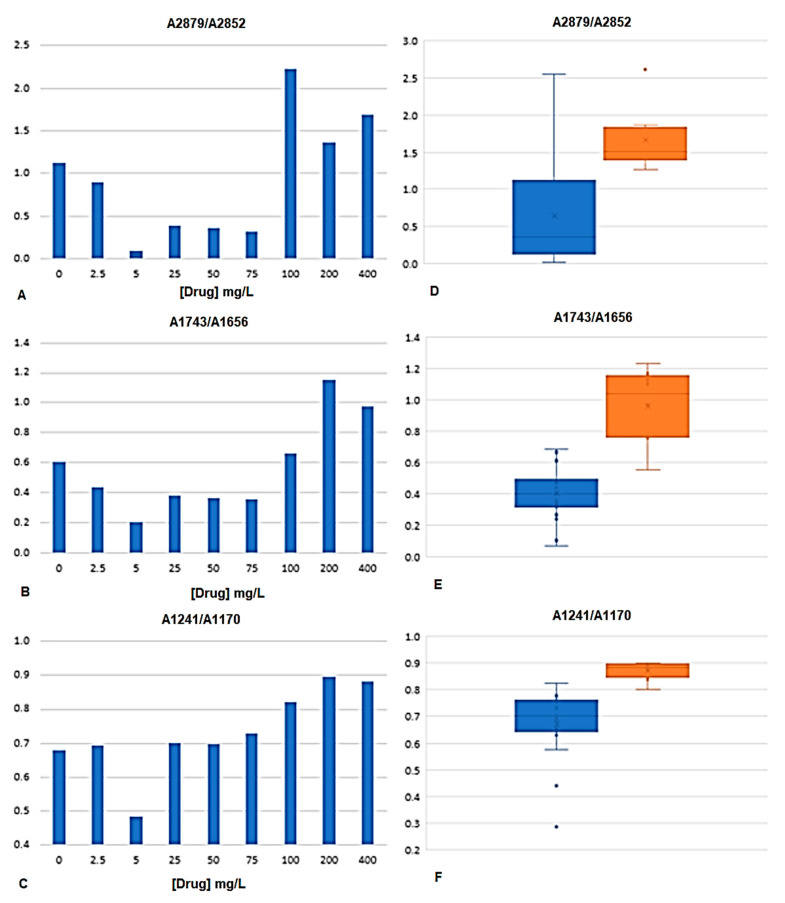
Peak ratios of bacteria spectra at A2879/A2852, A1743/A1656, and A1241/A1170. **(A–C)** Average values obtained along with diverse drugs concentration. **(D–F)** Box-plot graphs of peak ratios from drug concentrations lower to the MIC (blue boxes) and equal or higher to the MIC (orange boxes).

**Figure 6 antibiotics-09-00897-f006:**
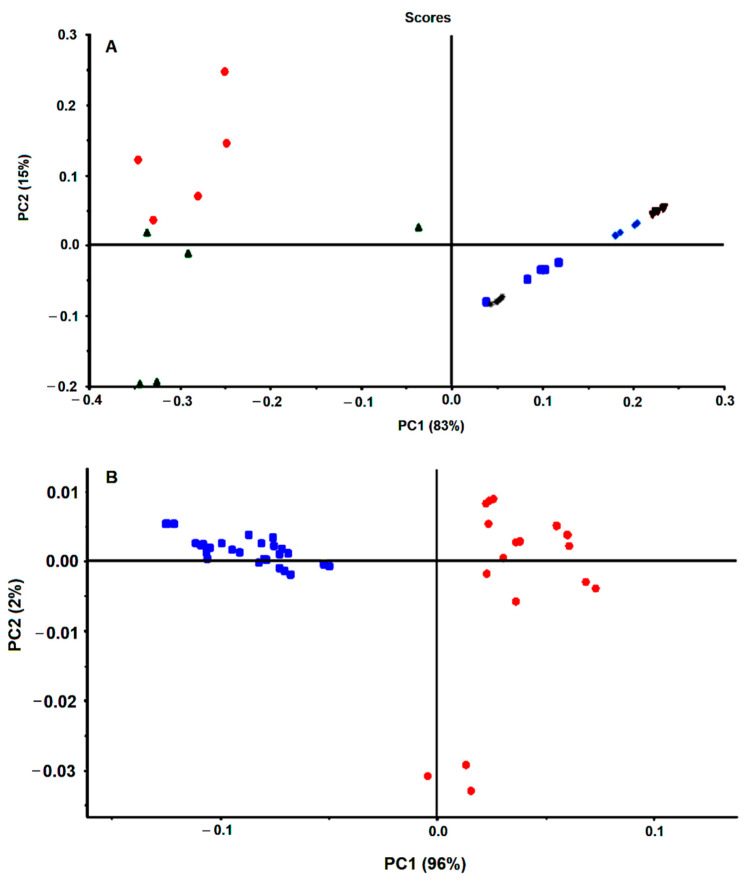
PCA of first derivative spectra of *H. pylori* strains. (**A**) 26,695 *H. pylori* (blue) and J99 (red) incubated or not with antimicrobials. (**B**) Six *H. pylori* strains not incubated with drugs: Blue, 207-99; Red, 939-99; Green, B1-99; Light blue, 927-03; Brown, 1152-04; Grey, 1198-04.

**Figure 7 antibiotics-09-00897-f007:**
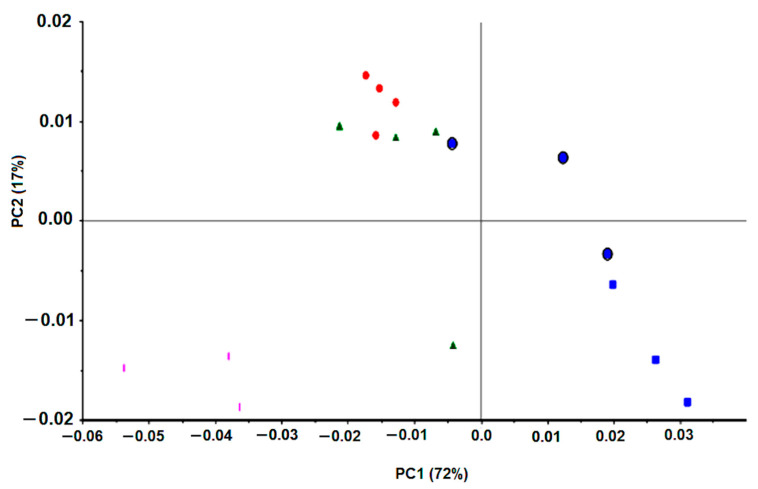
PCA of first derivative spectra of J99 *H. pylori* incubated with different clarithromycin concentrations: Blue square, 0 mg/L; Blue circle, 1.5 to 7.5 mg/L; Red, 15 mg/L (MIC); Green, 30 mg/L; Pink, 75 mg/L.

**Figure 8 antibiotics-09-00897-f008:**
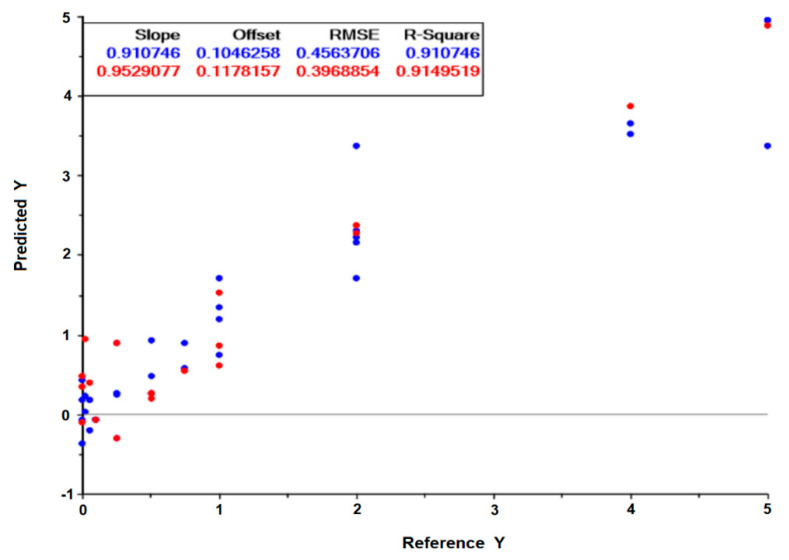
Predicted drug concentration in relation to the experimental drug concentration, obtained from the PLS regression model built on 7 latent variables. The incubation of the reference 26,695 and J99 *H. pylori* strains with the peptide-based antimicrobial and clarithromycin, respectively, were considered. The graph scales are relative to the MIC of the drug. Blue and red symbols are relative to the calibration and validation model, respectively.

**Table 1 antibiotics-09-00897-t001:** Composition of the incubation mixture, containing *H. pylori* 26,695 strain, brucella broth, and a peptide-based drug. The incubation period (at 37 °C) of the mixture is also indicated. The experiments were conducted in triplicate.

Experiment Number	*H. pylori* Quantity (O.D.600 nm)	Brucella Broth (% (*w/v*))	Concentration of the Drug Model (mg/L)	Incubation Time (h)
1	4.0	25	0, 2.5, 25, 50, 75, 100	1
2	2.0	5	0, 2.5, 5, 25, 50, 75, 100, 200, 400	1
3	2.0	5	0, 2.5, 5, 25, 50, 75, 100, 200, 400	3
4	2.0	5	0, 2.5, 5, 25, 50, 75, 100, 200, 400	6

**Table 2 antibiotics-09-00897-t002:** Variance within the spectra triplicates of each experimental condition within a spectra PCA, of *H. pylori* treated with different concentrations of the drug model.

Drug Model Concentration (Mg/L)	Variance in the Triplicates of Data Set Prior to MSC Correction	Variance in the Triplicates of Data Set after MSC Correction
0	1.52	0.44
2.5	0.79	0.17
25	5.44	0.78
50	3.62	0.71
75	3.08	0.77
100	8.97	1.94

**Table 3 antibiotics-09-00897-t003:** Average values and standard deviation of ratios of peak heights from the bacteria spectra and the *p*-value from ANOVA between the incubation of 26,695 *H. pylori* with concentrations of the peptide-based drug lower than the MIC and bacteria incubated with drugs concentrations higher or equal to the MIC.

Peak Ratio	A-Bacteria Incubated with [Drugs] < MIC Average SD		B-Bacteria Incubated with [Drugs] ≥ MIC Average SD		*p*-Value of ANOVA between “A” and “B” at 5% Significance
A3301/A3064	6.00	4.19	3.39	6.93	not significant
A3301/A2921	1.54	43.69	0.44	106.03	not significant
A2921/A2852	3.13	1.85	1.38	3.04	not significant
A2879/A2852	0.53	1.76	0.62	0.78	0.0022
A2960/A2921	0.64	84.47	0.32	213.31	not significant
A1743/A1656	0.39	0.93	0.17	0.34	3.7 × 10^−7^
A1743/A1548	1.29	34.52	0.85	42.86	0.0003
A1656/A1548	3.01	31.41	0.78	37.39	0.0016
A1548/A1500	12.29	0.18	44.44	0.75	not significant
A1457/A1407	1.07	1.11	0.08	0.02	not significant
A1457/A1548	1.06	29.99	0.72	37.00	0.0011
A1241/A1079	1.91	2.95	1.66	2.59	not significant
A1241/A1170	0.66	0.87	0.12	0.07	0.0005
A1241/A1548	0.76	20.83	0.41	25.59	0.0011
A1033/A1079	2.42	2.33	1.54	1.51	no significant

**Table 4 antibiotics-09-00897-t004:** Average values and standard deviation of peak ratios from the J99 *H. pylori* spectra and the *p*-value at 5% significance from ANOVA between the incubations with clarithromycin concentrations lower or higher to the MIC.

Peak Ratio	A-Bacteria Incubated with [Drugs] < MIC Average SD		B-Bacteria Incubated with [Drugs] ≥ MIC Average SD		*p*-Value of ANOVA between “A” and “B” at 5% Significance
A3301/A3064	2.84	0.07	2.86	0.19	not significant
A3301/A2921	2.41	0.03	2.39	0.14	not significant
A2921/A2852	2.06	0.02	2.06	0.07	not significant
A2879/A2852	0.93	0.01	0.93	0.02	not significant
A2960/A2921	0.90	0.01	0.89	0.02	not significant
A1743/A1656	0.05	0.00	0.05	0.00	not significant
A1743/A1548	0.10	0.00	0.10	0.01	not significant
A1656/A1548	1.89	0.01	1.81	0.09	0.0014
A1548/A1500	4.02	0.12	3.67	0.53	0.015
A1457/A1407	0.62	0.04	0.66	0.06	0.020
A1457/A1548	0.16	0.01	0.18	0.04	0.046
A1241/A1079	0.46	0.02	0.51	0.03	0.0000047
A1241/A1170	2.85	0.11	3.35	0.37	0.000022
A1241/A1548	0.19	0.00	0.20	0.01	0.0019
A1033/A1079	0.33	0.01	0.37	0.06	0.0091

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
