# Peer review of "Potential of FTIR-Spectroscopy for Drugs Screening against Helicobacter pylori"

_antibiotics, 2020, doi:10.3390/antibiotics9120897_

Round 1
Reviewer 1 Report
In this manuscript the authors describe the potential application of FTIR-based analysis of drug MICs against H. pylori. This manuscript presents an early development of the idea with potential applications in laboratory. Although this technique provides some advantages but it cannot, at this stage, replace the standard MIC assay against the bacteria.
As this manuscript is a resubmitted one and the authors have mentioned clearly enough in this version of the manuscript about the potential and the limitation of this technique. Therefore it will be accepted once the author have again checked the grammar and sentences in the manuscript with an appropriate native language service.
Author Response
Subject: Resubmission of revised paper “Potential of FTIR-spectroscopy for drugs screening against Helicobacter pylori”
Reviewer 1
We have the pleasure to submit the final version of the manuscript. We appreciated and thanks to all the reviewer’s comments that strongly contributed for the manuscript improvement.

Reviewer 2 Report
I recommend a few modification.
Double check the English.
Can you reexplain the paragraph: 3.5. Chemometrics analysis, it's not clear written
Conclusion need to be short and clear. Please phrase 3 short conclusion.
Author Response
Subject: Resubmission of revised paper “Potential of FTIR-spectroscopy for drugs screening against Helicobacter pylori”
Reviewer 2
We have the pleasure to submit the final version of the manuscript. We appreciated and thanks to all the reviewer’s comments that strongly contributed for the manuscript improvement.
